# Superimposed Pulmonary Tuberculosis (PTB) in a 26-Year-Old Female with No Underlying Co-Morbidities Recovering from COVID-19—Case Report

**DOI:** 10.3390/tropicalmed8050268

**Published:** 2023-05-08

**Authors:** Katanekwa Njekwa, Monde Muyoyeta, Bavin Mulenga, Caroline Cleopatra Chisenga, Michelo Simuyandi, Roma Chilengi

**Affiliations:** 1Center for Infectious Disease Research in Zambia (CIDRZ), P.O. Box 34681, Lusaka 10101, Zambia; monde.muyoyeta@cidrz.org (M.M.); bavin.mulenga@cidrz.org (B.M.); caroline.chisenga@cidrz.org (C.C.C.); michelo.simuyandi@cidrz.org (M.S.); roma.chilengi@cidrz.org (R.C.); 2Tuberculosis Programs—Director, Centre for Infectious Disease Research, P.O. Box 34681, Lusaka 10101, Zambia; 3Enteric Diseases and Vaccine Research Unit (EDVRU)—Director, P.O. Box 34681, Lusaka 10101, Zambia; 4Zambia National Public Health Institute (ZNPHI)—Director, H9M2+WGX, Lusaka 10101, Zambia

**Keywords:** tuberculosis (TB), COVID-19, SARS-CoV-2, rt-PCR

## Abstract

Tuberculosis before the COVID-19 pandemic is said to have killed more people globally than any other communicable disease and is ranked the 13th cause of death, according to the WHO. Tuberculosis also still remains highly endemic, especially in LIMCs with a high burden of people living with HIV/AIDS, in which it is the leading cause of mortality. Given the risk factors associated with COVID-19, the cross similarities between tuberculosis and COVID-19 symptoms, and the paucity of data on how both diseases impact each other, there is a need to generate more information on COVID-19–TB co-infection. In this case report, we present a young female patient of reproductive age with no underlying comorbidities recovering from COVID-19, who later presented with pulmonary tuberculosis. It describes the series of investigations performed and treatments given during the follow-up. There is a need for more surveillance for possible COVID-19–TB co-infection cases and further research to understand the impact of COVID-19 on tuberculosis and vice versa, especially in LMICs.

## 1. Introduction

SARS-CoV-2 is the cause of coronavirus disease 2019 (COVID-19), a kind of viral pneumonia [1,2]. The World Health Organization (WHO) described COVID-19 as a pandemic on the 11 March 2020 [3], and it has remained a major global health concern to date. Globally, according to the WHO, there have been about 753,479,439 confirmed cases of COVID-19, including 6,812,798 deaths and a total of 13,156,047,747 vaccine doses administered [4]. Zambia, since the beginning of the COVID-19 pandemic, has had an approximate cumulative total of 340,582 confirmed cases, 2883 deaths, and about 8,668,658 COVID-19 full vaccinations [5]. 

The natural history of symptoms in COVID-19 patients is highly variable, ranging from asymptomatic mild infection (mainly in children or the young) to multi-organ failure, frequently resulting in death, which is mainly in the elderly [6] and patients with underlying co-morbidities [7,8]. COVID-19 pneumonia has been associated with high morbidity and mortality rates [9]. One of the major complications in patients hospitalized with COVID-19 is secondary bacterial infection of the respiratory system. *Mycobacterium tuberculosis* is the bacterium that causes tuberculosis, which usually attacks the lungs but can attack any part of the body [10,11,12,13]. It is estimated that one-quarter of the world’s population is latently infected with TB and has a 5–10% lifetime risk of falling ill with TB, and those with compromised immune systems, such as people living with HIV, malnutrition, or diabetes, or people who use tobacco are considered to be at high risk [14]. During the COVID-19 pandemic, tuberculosis became the world’s second deadliest infectious killer after the SARS-CoV-2 virus [15]. COVID-19 and TB are two major infectious diseases posing significant public health threats, and co-infection results in complicated clinical conditions; there has been little research in this area [16]. An indicator that has been suggested to assess the impact of COVID-related disruptions on essential TB services at global, regional, and country levels is the number of notifications of people diagnosed with TB [17]. Worldwide, it is estimated that SARS-CoV-2 and *M. tuberculosis* together may have caused approximately 5.7 million deaths in the past 2 years [18]. This case report describes an event of COVID-19–PTB co-infection in a young female patient. 

## 2. Case Representation

A 26-year-old female, who was a nurse by profession, tested positive for COVID-19 by RT-PCR (NPS), which was collected on the 31 May 2021. Prior to this, the patient had a history of general body weakness, headache, and chills for some days. The patient also experienced some episodes of vomiting 3 days after the COVID-19 test. On examination, she was clinically stable with no significant clinical findings and was placed under the community model management for COVID-19 patients, in accordance with the Zambian treatment guidelines. She was commenced on azithromycin, cephalexin, paracetamol, and chlorpheniramine oral medications. Four serial COVID-19 tests were performed on 31 May 2021, 9 June 2021, 18 June 2021, and 23 June 2021, respectively (Figure 1 and Table 1). The first three results were positive, while the fourth one was negative.

The patient did not recover clinically, and by mid-July 2021, she reported a one-week history of a headache, followed by fevers and body chills for four days. The highest temperature measured was 40 °C. Night sweats were present for about a week. The patient lost her appetite and had some noted weight reduction from 48 kg to 45 kg. The patient had joint pains for 1 week, which were relieved by diclofenac. During this period, the patient also experienced a productive, non-bloody cough, which was worsened by exposure to dust. The patient had generalized chest pains, which were described as twisting and was associated with left back pain. Difficulties with breathing were present, especially when laying down, though the patient could not give the actual duration.

The patient, post negative COVID-19 test, continued to present with a mild productive cough, but later, when her symptoms worsened as described as above, she presented to the local clinic, where a chest X-ray was ordered. On 24 July 2021, the patient was diagnosed with pulmonary TB, based on an X-ray (Figure 2), and two days later, the sputum result was positive for TB on GeneXpert MTB RIF (Cepheid) ultra; she was initiated on the four-drug regimen: isoniazid, rifampin, pyrazinamide, and ethambutol. She was also started on vitamin B_6_, prednisolone, and erythromycin on the same day at the local clinic. The patient later complained of severe episodes of nausea and vomiting which necessitated the withdrawal of erythromycin on suspicion of toxicity and end-organ damage. The series of investigations performed during the period of the patient’s illness are depicted in Figure 1 and Table 1 below. 

The patient was HIV negative and was diagnosed with COVID-19 for the first time; there was neither a history of TB disease nor close contact with a known TB patient. She had no known underlying medical conditions. She lived with five other family members in a four-roomed house reported to be well ventilated. The patient was single, not pregnant, and had no children, and she had no history of smoking, taking alcohol, or substance abuse.

After the PTB diagnosis, the patient was managed as an outpatient because she was clinically stable except for a few mild symptoms, which included reduced tolerance to exercise, mild cough, and mild chest discomfort, and fevers were reduced subsequently. Given her history of COVID-19, she was followed up closely, and she was to report to the clinic if there was any change in symptoms. Blood for routine ATT monitoring was collected (Figure 1 and Table 1). There was no initial chest X-ray taken during her COVID-19 infection to compare with the PTB chest X-ray. A repeat check X-ray was then scheduled while she was on ATT. The patient later reported symptom resolution, resumed work, and continued taking her ATT.

A repeat check chest X-ray shown on the right above Figure 2 reported an opacified left lower zone in addition to the patchy opacification/nodularity of the left mid and right upper zones, which could, in addition to lymphadenopathy, represent an ongoing PTB. There was no significant interval change since the first imaging. A series of blood tests were conducted, as summarized in Table 1.

## 3. Discussion

### 3.1. COVID-19 and PTB

In this report, we describe the case of a young female patient of reproductive age who presented with PTB after recovering from COVID-19. The unfolding of events through the course of the infection is described in Figure 1; this includes the times of symptomatic COVID-19, investigations, management, and the clinical outcome. Given the diverse clinical presentation of COVID-19 and the lack of focused medical treatment [19], the COVID-19 pandemic has added an additional burden to already overstretched health systems in SSA, which, among other things, have been focused on the longstanding dual epidemics of TB and HIV [20]. Zambia is listed among the 30 countries with a high burden of TB, HIV-associated TB, and resistant TB (MDR/RR-TB) [21]. In 2020 alone, Zambia was estimated to have a TB incidence of 319/100,000, and 39% of people with TB were also HIV-infected [22]. The TB and COVID-19 clinical symptoms, risk factors, and modes of transmission often tend to be similar [18,23,24], and because of this similarity in clinical characteristics, diagnostic difficulties arise, which may contribute to the development of severe COVID-19 [25]. The impact of COVID-19 on TB and vice versa is yet to be fully understood. In this patient, there could have been latent PTB; the possibility of it predisposing the patient to COVID-19 cannot be entirely ruled out. Some evidence of a slow progressive imbalance and deregulation of inflammatory markers in latent TB and understanding the pathogenesis of COVID-19/TB individually are crucial to understanding their roles in co-infection and how LTBI may become activated [26]. It has been suggested that COVID-19 can occur at any time during a patient’s TB cycle, with worse outcomes in pulmonary TB disease co-infection, but more evidence is needed to understand the potential of COVID-19 to favor the reactivation of an existing TB infection [27]. Pre-existing lung diseases may be important predictors for mortality and severe disease outcomes in COVID-19 patients [28,29,30]. Previous lung disease, such as treated or untreated TB, and old age are independent risk factors of a worse prognosis for those infected with COVID-19 [31]. On the other hand, SARS-CoV-2 infection may also markedly affect and challenge the immune system by causing leukopenia, lymphopenia, and an inflammatory cytokine storm, due to immunological changes [32,33]. As SARS-CoV-2 infection causes the blockade of the innate immunity and spreads from the upper airways to the alveoli in the early phases, it can replicate with no local resistance, causing pneumonia [34]. There is a need to understand the cellular populations, functional pathways, and signatures of the immunopathogenesis of TB, as they pose a dual risk to COVID-19 severity and TB disease progression. Dysregulation of T cells in COVID-19 may increase disease severity, and a reduction in Mtb-specific CD4+ T cells, may have possible implications for TB disease progression [35,36,37,38,39]. In post-mortems performed on 21 sudden and unexpected COVID-19 community and hospital deaths in Zambia, 2 and 1 were caused by PTB, respectively [40]. There were some independently reported cases of confirmed Mycobacterium tuberculosis infection prior to COVID-19, COVID-19–PTB, and COVID-19–PTB–HIV co-infections [41,42,43,44,45]. 

### 3.2. Clinical Symptoms

Though both TB and COVID-19 primarily involve the lungs and have somewhat similar symptoms e.g., cough, fever, and difficulty breathing, clinical features differ in certain aspects. COVID-19 has a much more rapid onset and an incubation period of about one to two weeks, while the clinical manifestations of TB typically develop over a much longer period [24]. As SARS-CoV-2 continues to evolve, the clinical symptomatology of COVID-19 is proving to be extremely variable across different individual, demographic, and geological levels [46]. This patient presented with milder symptoms of COVID-19, recovered, and about 1 month later, developed PTB (Figure 1). With reference to the high endemic nature of TB and TB exposure, as well as given the similarity in their transmissibility routes, it can be argued that massive screening for both diseases should be performed. This also means that if the COVID-19 safety guidelines are upheld, the spread of TB will also be minimized. What also adds to many scientific questions in this case is the age of the patient (young), the patient not being pregnant (Table 1), and the fact that the patient did not have any underlying co-morbidities. Both TB and COVID-19 have some similarities in the socio-demographic and medical background risk profiles of patients. COVID-19 mortalities are seen more in older patients who have more complications associated with hypertension, hepatitis, and cancer and more symptoms of dyspnea. They are also more likely to have CT imaging features of bilateral lesions, infiltrates, tree-in-bud, and higher leucocyte counts than survivors [47].

### 3.3. Radiological Findings

Pulmonary tuberculosis (PTB), depending on the extent of active infection, may manifest as lymphadenopathy, pulmonary consolidation, cavities, centrilobular nodules, miliary lung nodules, and pleural effusion [48,49]. Chest X-rays may be central in the screening and diagnosis of PTB, but sputum culture remains a gold standard for diagnosis [50,51]. In this patient, the chest symptoms post negative COVID-19 test increased, and the findings on the chest X-ray suggestive of PTB (Figure 2) were incidental findings but were confirmed by the positive Mycobacterium tuberculosis in sputum. Chest X-rays have a poor specificity; although some abnormalities are rather specific for pulmonary TB (for example, cavities), many chest X-ray abnormalities that are consistent with pulmonary TB are seen in several other lung pathologies [52]. There was no chest X-ray ordered during the period when the patient had active COVID-19, either due to her not having enough pulmonary symptoms to warrant the investigation or due to some research findings suggesting that chest radiography is regarded as not sufficiently informative by clinicians in a COVID-19 pandemic triage setting [53], even though the chest X-ray is a useful tool for detecting changes to suggest the diagnosis. The CT of the chest, however, has a higher sensitivity [54]. Another reason why chest X-rays may not be ordered in COVID-19 is the risk of cross infection, given the nature of SARS-CoV-2. In this patient, the repeat chest X-ray (Figure 2), according to the radiographer’s report, showed persistence in radiological findings, compared to the first chest X-ray (Figure 2); chest X-rays, on the other hand, may play an important role in the follow-up of patients with a history of COVID-19 pneumonia [55].

### 3.4. Laboratory Findings

During the course of COVID-19 infection, there may be subtle hematological changes that might appear early; significant hematological changes associated with progressive disease may guide the management and, in some cases, predict the outcome [56]. Temporal dynamic changes in COVID-19 patients, such as lymphocyte count/percentage, neutrophil count/percentage, lymphocyte–monocyte ratio (LMR) and neutrophil-to-lymphocyte ratio (NLR), and platelet count in the blood are seen to be different between survivors and non-survivors [32,57,58]. Some of the other common hematological abnormalities of COVID-19 include lymphopenia, thrombocytopenia, and elevated D-dimer levels [59]. Serial blood tests (Figure 1 and Table 1) were performed on this patient through the course of the disease; the gradual increase in eosinophilic % and basophil % were noted, as well as the absolute count of monocytes and occasional increases in absolute counts of neutrophils and basophils. The gradual decrease in hemoglobin and the thrombocytosis were also noted. Whether or not the FBC picture was a response to the COVID-19 infection or PTB would require more investigations in a larger number of patients with similar presentations as this patient. In PTB, some of the hematological parameters, such as the erythrocytic sedimentation rate (ESR), platelets, and leukocytes, work as hallmarks and help the clinicians in the early diagnosis of the disease [60,61]. Other predictors of tuberculosis are shown to be NLR > 1.19 and MLR > 0.29, which may be used to discriminate patients with bacterial CAP from patients with pulmonary TB [62,63]. How these subtle laboratory hematological changes implicate the clear distinction in predicting or differentiating COVID-19 or PTB is an area not yet fully explored but may play a crucial role in places with high burdens of TB. The other biomarker tests performed did not show any signs of end-organ damage in this patient. In this patient, the GeneXpert MTB RIF (Cepheid) ultra was used in the diagnosis of PTB. GeneXpert MTB RIF (Cepheid) ultra is the recommended diagnostic test in Zambia and is the next generation after the Xpert MTB RIF of 2010. One sample that tests positive by GeneXpert is diagnostic of TB. Diagnosis of TB in this patient followed the diagnosis guidelines of Zambia and the global diagnostic guidelines for TB. It was suggested that despite the Xpert Ultra’s improved sensitivity, it has a 2% lower specificity than Xpert (96% (94–97) vs. 98% (97–99)), and that the specificity of Xpert Ultra for the detection of Mycobacterium tuberculosis is even lower in patients with previous tuberculosis or in patients from high incidence countries [64].

After the initial positive SARS-CoV-2 test, patients may continue to test positive on antigen tests for a few weeks and on NAATs for up to 90 days [65]. The patient had a series of COVID-19 tests that included three PCRs and one antigen test. The first three positive COVID-19 tests were over a period of more than fourteen days, with the first, second, and third at day one, day ten, and day eighteen, respectively. A negative test was recorded at about 23 days from the initial positive COVID-19 test. Persistent COVID-19 can be defined as active viral replication beyond the currently accepted post-exposure quarantine period of 14 days [66]. It was suggested that viral shedding may occur following recovery but does not appear to play a role in transmission in relatively healthy people after more than 10 days following the onset of infection [67].

## 4. Conclusions

The co-infection of SARS-CoV-2 and Mycobacterium tuberculosis is still not yet fully understood. So far, what is known is that the clinical symptoms are similar, and clearly differentiating them may prove to be a challenge for management and ultimately affect patient outcome. Therefore, clinicians working in high TB-endemic areas should take COVID–TB co-infection into consideration, especially in situations where respiratory symptoms do not resolve, instead of just focusing on one disease. This patient did not fit the typical patient risk profile of both COVID-19 and PTB, but as the COVID-19 pandemic continues to evolve, there is an urgent need to keep updating the scientific body with such rare cases.

The full impact of COVID-19 on TB surveillance is yet to be understood; according to the WHO, it is critical that TB services are not disrupted during the COVID-19 response [24]. On the other hand, a study in Zambia showed that the COVID-19 pandemic did not seem to deter patients from care-seeking for TB [68]. It was suggested to routinely screen for M. tuberculosis among suspected or confirmed cases of COVID-19 in high TB-burden countries, due to the worse prognosis of COVID–TB and the confounding clinical symptoms of these two diseases [23,48]. In conclusion, given the limited information on COVID-19–TB co-infection, there is an urgent need for larger prospective studies to investigate the timely diagnosis, targeted management, follow-up (long-COVID-19 or COVID-19 hauler), and clear clinical course of COVID-19 patients with active and/or latent TB.

## Figures and Tables

**Figure 1 tropicalmed-08-00268-f001:**
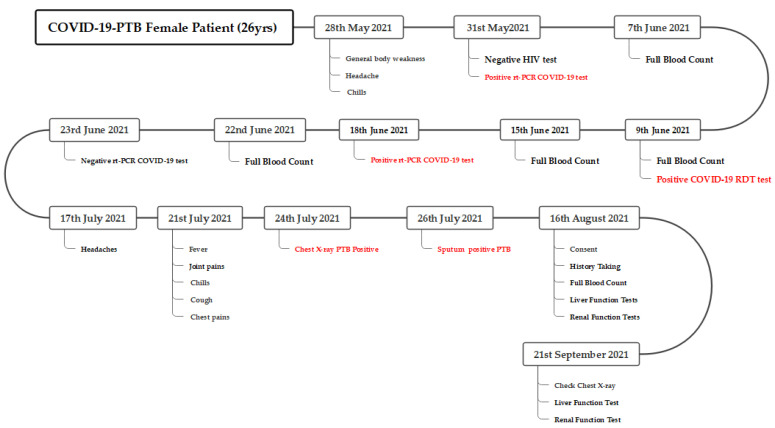
The figure above highlights the unfolding of events, with specific descriptions of the timelines of events.

**Figure 2 tropicalmed-08-00268-f002:**
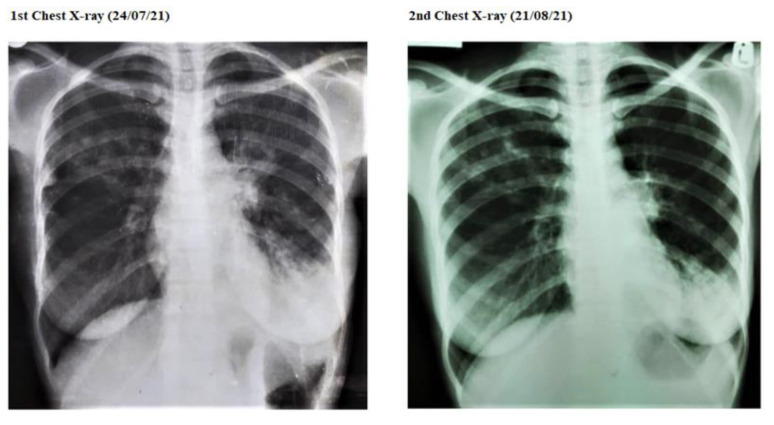
First and second chest X-ray images (radiographer’s report comparison of the two images). Persistent opacified/consolidative left lower zone in addition to patchy opacification of the left mid and right upper zones. Persistent nodularity within the right upper zone in addition to fissure thickening is evident. Remainder of the lungs are clear. Costo-phrenic angles are not blunted. Persistent left hilar fullness suspicious for adenopathy. Heart size is normal. Cardio-pericardial silhouette is normal.

**Table 1 tropicalmed-08-00268-t001:** The table below is a summary of laboratory results for blood. Full blood count (FBC), chemistry (LFTs, RFTs), COVID-19 tests, TB tests, and pregnancy tests.

Dates	31 May 2021	7 June 2021	9 June 2021	15 June 2021	18 June 2021	22 June 2021	23 June 2021	26 July 2021	16 August 2021	21 September 2021	Ref. Range
**Hematology**											
**Complete blood count**											
White Blood Cell		6.46	7.02	**7.76**		**8.13 ꜛ**			5.78	5.89	×10^9^/L (2.96–7.58)
Red Cell Count		4.49	4.43	4.34		4.44			4.24	4.11	×10^12^/L (4.04–5.52)
Hemoglobin		12	11.9	11.8		12.1			**10.9 ꜜ**	**10.8 ꜜ**	g/dL (11–15.7)
Hematocrit		38.6	38.1	37.6		37.8			35.5	34.2	% (33.3–45)
MCV		86	86.0	86.6		85.1			83.7	83.2	fL(71.4–94)
MCH		26.7	26.9	27.2		27.3			25.7	26.3	Pg (22.7–33.3)
MCHC		31.1	31.2	31.4		32.0			**30.7 ꜜ**	31.6	g/dL (30.9–36.5)
Platelet Count		408	369	**418 ꜛ**		**615 ꜛ**			**438 ꜛ**	**460 ꜛ**	×10^9^/L (156–411)
**Differential Count**											
Neutrophils#		4.07	3.63	4.41		**5.06 ꜛ**			3	2.45	×10^9^/L (1.01–4.49)
Lymphocytes#		1.88	2.39	2.46		2.24			1.75	2.39	×10^9^/L (1.11–3.12)
Monocytes#		0.36	**0.66 ꜛ**	**0.71 ꜛ**		**0.67 ꜛ**			**0.83 ꜛ**	0.47	×10^9^/L (0.21–0.59)
Eosinophils#		0.12	0.30	0.15		0.12			0.15	**0.58 ꜛ**	×10^9^/L (0.01–0.39)
Basophil#		0.03	0.04	0.03		0.04			0.05	**5.89 ꜛ**	×10^9^/L (0.00–0.05)
Neutrophils%		62.9	51.7	56.9		62.2			51.8	41.6	% (28.5–70.8)
Lymphocytes%		29.1	34.0	31.7		27.6			30.3	40.6	% (21.4–58.8)
Monocytes%		5.6	9.4	9.1		8.2			**14.4 ꜛ**	8.0	% (4.6–12.2)
Eosinophils%		**1.9 ꜛ**	**4.3 ꜛ**	**1.9 ꜛ**		**1.5 ꜛ**			**2.6 ꜛ**	**9.8 ꜛ**	% (0.2–0.8)
Basophils%		0.5	0.6	0.4		0.5			0.9	**100 ꜛ**	% (0.0–1.0)
**Liver Function Tests**											
Aspartate Transferase									28	34	U/L (0–38)
Alanine Transferase									22	11	U/L (0–33)
Alkaline Phosphatase									84	58	U/L (30–120)
GGT										15.4	U/L (8.0–51.0)
Total Bilirubin										**3 ꜜ**	umol/L (6.5–27.0)
Direct Bilirubin										1.21	umol/L (0.00–6.24)
**Renal Function Tests**											
Blood Urea Nitrogen									2.6		mmol/L (0.0–8.3)
Creatinine									**57.6 ꜜ**		umol/L (60.0–112.0)
HIV Test	**Negative**										
**COVID-19 Tests**											
RT-PCR	**Positive**										
Antigen			**Positive**								
RT-PCR					**Positive**						
RT-PCR							**Negative**				
**PTB Test**											
Sputum Gene X-pert								**Positive**			
**Pregnancy Test**											
Urine		**Negative**									

ꜜꜛ Results in bold with a superscript are indicative of a change in laboratory value from the normal range.

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
