# Peer review of "Superimposed Pulmonary Tuberculosis (PTB) in a 26-Year-Old Female with No Underlying Co-Morbidities Recovering from COVID-19—Case Report"

_tropicalmed, 2023, doi:10.3390/tropicalmed8050268_

Round 1
Reviewer 1 Report
The manuscript is about presenting a case study of a covid infected person in whom TB was also diagnosed. The authors have reported all the laboratory, clinical examination reports and has presented the time series of the investigations too. But the results does not pinpoint or reveal any important significant indicator to dwell upon rather have quoted many of the referenced results only, but the same findings could not be found in this particular case study. The X-ray findings at two different time periods did not add any thing more to the conclusion and similarly the subtle changes as the authors have mentioned in case of the laboratory investigations also lead to no where and does not predict any thing significant nor does not support the conclusion.
Author Response
Dear Reviewer 1,
Thank you for taking the time to review the case report.
I have attached the word documents with the responses to the comments for your further review.
Please find attached.
Thank you again,
Katanekwa

Reviewer 2 Report
Superimposed Pulmonary Tuberculosis (PTB) in 26 years old female with no underlying co-morbidities recovering from COVID-19 - Case report.
The authors document COVID-19 infection in a nurse whose viral infection triggered latent pulmonary tuberculosis into active disease. This is an important and increasingly recognized outcome of the pandemic and one which could have important consequences in health-care settings as well as at-risk populations.
Main points.
1. Re: the GeneXpert TB diagnostic test. This test is mentioned only briefly in the diagnosis of tuberculosis in this case report. The diagnosis of tuberculosis in the health care worker relies largely on this one NAA test. I assume this is the first generation test of 2010? Authors, please include some further details of the test for the general reader. E.g. the manufacturer, the test variant and platform used; its limitations.
The test will detect pathogen DNA from both viable and non-viable mycobacterial genomes. It is not meant to replace conventional TB and MDR testing by growth on media but rather to give a fast indication of infection and rifampicin sensitivity. Therefore, were no routine methods, for example AFB smear / culture or IGRA assays also used to confirm the diagnosis and drug sensitivity and monitor progress of the patient whilst on ATT?
2. English grammar / Journal style.
The manuscript is generally well written and comprehensible but there are a number of places where references cited should be contained within the same, rather than separate, brackets and brought within the punctuation of the sentence to which they relate. There are numerous instances of this throughout the MS so that this rather unusual style is the norm.
For example lines 40-42 inclusive and onwards in the MS.
“…with COVID-19 is secondary bacterial infection of the respiratory system.[10],[11] Myco-40 bacterium tuberculosis is the bacterium that causes tuberculosis.[12] The bacteria usually 41 attacks the lungs, but TB bacteria can attack any part of the body.[13] It is estimated that…” and
line 141 “Pre-existing lung diseases may be important predictors for mortality and severe disease outcomes, in 142 COVID-19 patients.[28],[29],[30] Previous lung disease such as treated or untreated TB …”
3. Figure 2. The trachea and main stem bronchi are mentioned in the caption but their relevance to the diagnosis is not.
4. Table 1. Many of the blood tests results have a superscripted symbol (†) but this is not explained - there is no Table caption in the pdf provided for peer review.
Minor points.
1. Line 86. Do the authors mean to say that at the time of the first covid-19 test the patient also tested negative for HIV? If so, this could be clarified and included in the flow diagram, Figure 1.
2. Line 136. latent TB (small cap).
3. Line 228. “Gotten” is rather slang for written English; suggest “obtained” or “recorded”.
Author Response
Dear Reviewer 2,
Thank you for taking the time to review the case report.
Please find attached the responses to the comments in the attached word document for your further review.
Thank you again,
Katanekwa
